# Physical Origin of the Dark Spot in the First Image of Supermassive Black Hole SgrA*

## Vyacheslav I. Dokuchaev

Institute for Nuclear Research of the Russian Academy of Sciences, 117312 Moscow, Russia; dokuchaev@inr.ac.ru

**Abstract:** We elucidate the physical origin of the dark spot in the image of supermassive black hole SgrA* presented very recently by the EHT collaboration. It is argued that this dark spot, which is noticeably smaller than the classical black hole shadow, is the northern hemisphere of the event horizon globe. The classical black hole shadow is unseen in the image of SgrA*. The dark spot in the image of SgrA* is projected within the position of the classical black hole shadow on the celestial sphere. The outer boundary of this dark spot is an equator on the event horizon globe.

**Keywords:** general relativity; black holes; cosmology; modified gravity

## 1. Introduction

Recent breakthrough observations of the SgrA* image by the EHT collaboration directly demonstrate the existence of black holes in the Universe [1–6]. A natural question arises on the physical origin of the dark spot at the presented image.

There have been numerous numerical calculations using images (or silhouettes) of black holes highlighting very near the black hole event horizon by the luminous falling matter (see, e.g., some historical examples [7–10]). Strictly speaking, the event horizon of the black hole is not observable. However, the "reconstructed" dark silhouette of the event horizon could be. Reconstruction means the possibility to observe photons emitted by the falling accretion matter in the vicinity of the event horizon. The last photons emitted in the vicinity of the event horizon and detected by a distant telescope mark the outer boundary of the dark spot in the black hole image. In the thin accretion disk model, this outer boundary of the dark spot is the "reconstructed" image of the equator at the event horizon globe. The dark spot in the image of SgrA* is the northern hemisphere at the black hole event horizon globe if the accretion disk is thin and the rotation axis of the black hole coincides with the Milky Way's rotation axis. At the same time, in the case of M87* the dark spot is the southern hemisphere of the black hole event horizon globe.

The classical black hole shadow, which is a captured cross-section of photons in the black hole's gravitational field [11,12], has a size of the order of 50 micro-arcseconds in the case of SgrA*, but the distinctive dark spot in the EHT image is notably smaller. It was explained earlier that this small dark spot is a lensed image of the black hole event horizon itself if there is highly luminous accreting matter in the very vicinity of the black hole event horizon [13–21].

The physical conditions for the luminous accreting matter in the very vicinity of the black hole event horizon are naturally realized in the Blandford—Znajek process [22] and were confirmed by recent general relativistic magnetohydrodynamic (GRMHD) simulations [23]. These simulations demonstrated clearly that an unsteady accretion disk is mainly geometrically thin at the very vicinity of the black hole event horizon. Based on these GRMHD simulations, we used here the geometrically thin accretion disk model for the interpretation of the SgrA* image.

## 2. Basic Equations

To describe the physical origin of the observed dark spot, it is supposed that the accretion disk around the supermassive black hole SgrA* is thin and the rotation axis of SgrA* coincides with the rotation axis of the Milky Way galaxy. The corresponding angular diameter of SgrA*'s shadow, $d_{sh} \approx 50$ µas, coincides approximately with the observed emission ring diameter [1]. The angular gravitational radius of SgrA* is approximately ten times smaller than $d_{sh}$.

We describe the gravitational field of the black hole in the framework of Einsteinian gravity by using the classical Kerr metric [12,24–32] describing the rotating black hole with a mass $M$ and with a black-hole-specific angular momentum (spin) $a = J/M$ in standard Boyer–Lindquist coordinates $(t, r, \theta, \phi)$ [25]:

$$ds^2 = -e^{2\nu}dt^2 + e^{2\psi}(d\phi - \omega dt)^2 + e^{2\mu_1}dr^2 + e^{2\mu_2}d\theta^2, \tag{1}$$

$$e^{2\nu} = \frac{\Sigma\Delta}{A}, \quad e^{2\psi} = \frac{A\sin^2\theta}{\Sigma}, \quad e^{2\mu_1} = \frac{\Sigma}{\Delta}, \quad e^{2\mu_2} = \Sigma, \quad \omega = \frac{2Mar}{A}, \tag{2}$$

$$\Delta = r^2 - 2Mr + a^2, \quad \Sigma = r^2 + a^2\cos^2\theta, \quad A = (r^2 + a^2)^2 - a^2\Delta\sin^2\theta. \tag{3}$$

$\omega$ is a specific frame-dragging angular velocity. It uses the evident units with the gravitational constant $G = 1$ and the velocity of light $c = 1$. For a further simplification of formulas, we also used the dimensional values for the black hole spin $a = J/M^2 \leq 1$, for space distances $r \Rightarrow r/M$, for time intervals $t \Rightarrow t/M$, etc. The black hole event horizon radius $r_h$ in the Kerr metric is the largest root of the quadratic equation $\Delta = 0$:

$$r_h = 1 + \sqrt{1 - a^2}, \tag{4}$$

It was demonstrated by Brandon Carter [26] that in the Kerr metric there are four integrals of motion for test particles: $\mu$—test particle mass, $E$—particle total energy, $L$—particle azimuth angular momentum and $Q$—Carter constant, which are related to the non-equatorial motion. The resulting first-order differential equations of motion for the test particle in the Kerr metric are [12,26–32]:

$$\Sigma\frac{dr}{d\tau} = \pm\sqrt{R(r)}, \tag{5}$$

$$\Sigma\frac{d\theta}{d\tau} = \pm\sqrt{\Theta(\theta)}, \tag{6}$$

$$\Sigma\frac{d\phi}{d\tau} = L\sin^{-2}\theta + a(\Delta^{-1}P - E), \tag{7}$$

$$\Sigma\frac{dt}{d\tau} = a(L - aE\sin^2\theta) + (r^2 + a^2)\Delta^{-1}P. \tag{8}$$

In these equations: $\tau$—the proper particle time or affine parameter along the trajectory of a massless ($\mu = 0$) particle. The effective radial potential $R(r)$ governs the radial motion of test particles:

$$R(r) = P^2 - \Delta[\mu^2 r^2 + (L - aE)^2 + Q], \tag{9}$$

where $P = E(r^2 + a^2) - aL$. Meantime, the effective polar potential $\Theta(\theta)$ governs the polar motion of test particles:

$$\Theta(\theta) = Q - \cos^2\theta[a^2(\mu^2 - E^2) + L^2\sin^{-2}\theta]. \tag{10}$$

The trajectories of massive test particles ($\mu \neq 0$) in the Kerr metric depend on three orbital parameters (constants of motion): $\gamma = E/\mu$, $\lambda = L/E$ and $q = \sqrt{Q}/E$. The trajectories of massless particles ($\mu \neq 0$) depend only on two parameters, $\lambda = L/E$ and $q = \sqrt{Q}/E$.

The corresponding integral forms for equations of motion are very useful for numerical calculations:

$$\oint \frac{dr}{\sqrt{R(r)}} = \oint \frac{d\theta}{\sqrt{\Theta(\theta)}}, \tag{11}$$

$$\tau = \oint \frac{r^2}{\sqrt{R(r)}} dr + \oint \frac{a^2 \cos^2 \theta}{\sqrt{\Theta(\theta)}} d\theta, \tag{12}$$

$$\phi = \oint \frac{aP}{\Delta \sqrt{R(r)}} dr + \oint \frac{L - aE \sin^2 \theta}{\sin^2 \theta \sqrt{\Theta(\theta)}} d\theta, \tag{13}$$

$$t = \oint \frac{(r^2 + a^2)P}{\Delta \sqrt{R(r)}} dr + \oint \frac{(L - aE \sin^2 \theta)a}{\sqrt{\Theta(\theta)}} d\theta, \tag{14}$$

where the effective potentials $R(r)$ and $\Theta(\theta)$ are defined in Equations (9) and (10). The integrals (11)–(14) are the so-called contour (path) integrals along the trajectory of test particle, which are monotonically growing along the test particle trajectory (i.e., without the changing of their signs in the radial and polar turning points).

The classical black hole shadow in the Kerr metric is defined in the parametric form $(\lambda, q) = (\lambda(r), q(r))$ [11,12]):

$$\lambda = \frac{(3 - r)r^2 - a^2(r + 1)}{a(r - 1)}, \quad q^2 = \frac{r^3[4a^2 - r(r - 3)^2]}{a^2(r - 1)^2}, \tag{15}$$

where $r$ is the radius of the so-called photon sphere. Parameters $\lambda$ and $q$ are, correspondingly, the horizontal and vertical impact parameters of photons on the celestial sphere for a distant observer placed in the black hole equatorial plane. A static distant observer, placed at the distant radius $r_0 \gg r_h$, at the given polar angle $\theta_0$ and at the given azimuth $\phi_0$, will see the incoming photons, the horizontal impact parameter $\alpha$ and vertical impact parameter $\beta$ on the celestial sphere [11,33,34]):

$$\alpha = -\frac{\lambda}{\sin \theta_0}, \quad \beta = \pm \sqrt{\Theta(\theta_0)}. \tag{16}$$

## 3. Physical Origin of the Dark Spot in the First Image of SgrA*

We describe the physical origin of the dark spot in the image of SgrA* by using numerical calculations of photon trajectories, starting a little bit above the black hole event horizon and reaching a distant static observer just above the black hole equatorial plane. Note that our interpretation of the dark spot in the image of SgrA* is based at the thin accretion disk model, predicted by the recent GRMHD simulations.

Figure 1 demonstrates the resulting composition of the observed SgrA* image with the dark silhouette in the northern hemisphere of the black hole event horizon in the thin accretion disk model. This dark region is projected on the celestial sphere inside the position of a classical black hole shadow (closed purple curve). The outer boundary of the dark region is an equator on the event horizon globe. The form of this boundary was numerically calculated by using the formalism described in the Section 2 for trajectories of photons emitted a little bit above the black hole event horizon and reaching a distant static observer. For more details, see

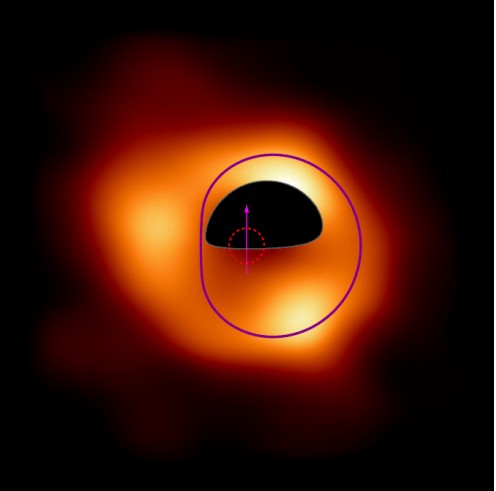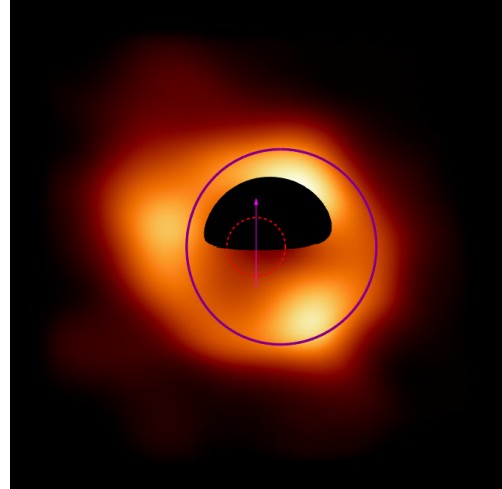

**Figure 1.** Composition of the observed image of SgrA* with the dark silhouette of the northern hemisphere of the black hole event horizon in the thin accretion disk model. This dark region is projected on the celestial sphere inside the position of classical black hole shadow (closed purple curve). The outer boundary of the dark region is an equator on the event horizon globe. The form of this boundary was numerically calculated by using trajectories of photons emitted a little bit above the black hole event horizon and reaching a distant static observer. Magenta arrows indicate the direction of the black hole's rotation axis. The dashed circles are the black hole event horizons in the imaginary Euclidean space without gravity. The left panel corresponds to the case of a black hole with the spin $a = 0.982$, and the right panel corresponds to the case of $a = 0.65$. For more details, see [13–21].

## 4. Conclusions

The aim of this note is the presentation of some evidence that the dark spot in the first image of the supermassive black hole SgrA* at the Milky Way's center, presented recently by the EHT collaboration, is the lensed image of the northern hemisphere of the event horizon globe if the luminous accretion disk around this black hole is geometrically thin. Namely, the geometrically thin accretion disk in the very vicinity of the black hole's event horizon was predicted by recent GRMHD simulations. Note that the classical black hole shadow (a captured cross-section of photons in the black hole's gravitational field) is definitely larger than the dark spot in the SgrA* image.

Black holes are the most amazing manifestation of general relativity (Einstein gravity) in the strong field limit. General relativity was verified experimentally only in the weak field limit inside the Solar System and the nearby galaxies before the direct EHT observations of supermassive black holes M87* and SgrA*. Nowadays, we are convinced that black holes are real astrophysical objects described by Einsteinian gravity in the strong field limit. The other manifestation of the strong field limit is in cosmology, where the use of Einsteinian gravity for interpretation of observational data begets the problems with enigmatic dark matter and dark energy. Meantime, we do not know that general relativity is a valid gravity theory because there are numerous modified gravity theories which also describe the black holes and may be used in cosmology. In the near future, the unique information for the verification or falsification of modified gravity theories will be provided by the detailed observations of black hole images with the projected Millimetron Space Observatory [35].

**Funding:** This research received no external funding.

**Informed Consent Statement:** Not applicable.

**Data Availability Statement:** Not applicable.

**Conflicts of Interest:** The author declares no conflict of interest.

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
