# Peer review of "Physical Origin of the Dark Spot in the First Image of Supermassive Black Hole SgrA*"

_2674-0346, doi:10.3390/astronomy1020009_

Round 1

Reviewer 1 Report

This paper discusses the physical horizon of the dark spot in the EHT observation of Sgr A*. Unfortunately, in my opinion, the claims of this paper are not particularly clear and, more importantly, not supported by convincing evidence. 

It is claimed in the introduction that the distinctive dark spot is smaller than the expected size. However, the size of the dark spot depends on the angular resolution, which is lowered by the variability of the source. This aspect is entirely ignored by the analysis reported in this paper.

The conclusions of this paper are not clear to me. In fact, with reference to Fig.~1, the purple curve does not match the dark spot. 

Furthermore, it is explained that the physical origin of the dark spot has been obtained ``by using numerical calculation of photon trajectories, starting a little bit above the black hole event horizon and reaching a distant static observer just above the black hole equatorial plane". This procedure is incorrect. In fact, most of the matter inside the photon sphere will not escape to infinity. This implies that the dark spot is actually associated with the photon sphere, not the horizon. 

Let me also stress that the paper assumes 
1. Thin accretion disk;
2. Rotation axis of SgrA* coincides with a rotation axis of the Milky Way galaxy.
Neither of these assumptions is justified in the text. 

Finally, let me point out a minor problem. The horizon is always called ``event horizon" rather than ``apparent horizon". This terminology is very common (also in the name of the EHT collaboration), but it is wrong anyway. The event horizon has a precise mathematical definition and it is not observable [arXiv:1407.7295]

Therefore, I am sorry to say that in my opinion this paper should not be published. 
Of course, if the author believes that I misunderstood the analysis, I am open to addressing their concerns and, eventually, referee the second version of this manuscript.

Author Response

Point 1: It is claimed in the introduction that the distinctive dark spot is smaller than the expected size. However, the size of the dark spot depends on the angular resolution, which is lowered by the variability of the source. This aspect is entirely ignored by the analysis reported in this paper.

Response 1: Angular resolution of the first SgrA* is very poor. For this reason the using of the transparent thin disk model in this note (or any other) is justified.

Point 2: The conclusions of this paper are not clear to me. In fact, with reference to Fig.~1, the purple curve does not match the dark spot.

Response 2: The purple curve does not match the dark spot because it is the classical black hole shadow, which is invisible in this image. This purple curve matches the luminous inner part of the accretion disk near the outer boundary of this purple curve. Instead of, namely the dark silhouette of the northern hemisphere of the black hole event horizon matches the rather small dark spot rather well (if black hole spin >0.5). In the revised note the Fig.1 is improved for better coincidence of the dark spot with our dark spot mode (by remembering the pure EHT angular resolution and 3-sigma uncertainty).

Point 3: Furthermore, it is explained that the physical origin of the dark spot has been obtained ``by using numerical calculation of photon trajectories, starting a little bit above the black hole event horizon and reaching a distant static observer just above the black hole equatorial plane". This procedure is incorrect. In fact, most of the matter inside the photon sphere will not escape to infinity. This implies that the dark spot is actually associated with the photon sphere, not the horizon.

Response 3: There is no any need for the matter inside the photon sphere to escape to infinity to provide the observable black hole image. Only photons from the lumunous accretion disk in the vicinity of the black hole event horizon may happily escape to a distant observer (with the correponding Doppler and gravitational energy shifts). It is verified in numerous published simulations and in our previous published papers.

Point 4: Let me also stress that the paper assumes 
1. Thin accretion disk;
2. Rotation axis of SgrA* coincides with a rotation axis of the Milky Way galaxy.
Neither of these assumptions is justified in the text.  

Response 4: In the revised note we state that ``thin accretion disk'' is a stated model in our note. It is a very natural idea (hypothesis) that ``rotation axis of SgrA* coincides with a rotation axis of the Milky Way galaxy'' proposed and used in many publications.

Point 5: Finally, let me point out a minor problem. The horizon is always called ``event horizon" rather than ``apparent horizon". This terminology is very common (also in the name of the EHT collaboration), but it is wrong anyway. The event horizon has a precise mathematical definition and it is not observable [arXiv:1407.7295].

Response 5: Yes, the event horizon of the black hole is not observable. However, the observable is the ``reconstructed’’ dark silhouette of the event horizon. Reconstruction means that it is possible to observe photons emitted be the infalling accretion matter in the vicinity of the event horizon. The last emitted photons and detected by a distant telescope mark the outer boundary of the dark spot at the black hole image. In the thin accretion disk model this outer boundary of the dark spot is the ``reconstructed’’ image of the equator at the black hole event horizon globe. In the thin accretion disk model the dark spot in the case of the SgrA* is the northern hemisphere at the black hole event horizon globe (while in the case of M87* the dark spot is the southern hemisphere at the black hole event horizon globe.). This clarification is added to Introduction of the updated note.

Reviewer 2 Report

Reviewer's comments

Manuscript details:
Journal: Astronomy
Manuscript ID: astronomy-1762982
Type of manuscript: Comment
Title: Physical origin of the dark spot at the first image of supermassive
black hole SgrA*
Authors: Vyacheslav Ivanovich Dokuchaev

Author discussed the physical origin of the dark spot (obtained by the EHT collaboration) at the image of SMBH SgrA*. He claim that the dark spot at the image of SgrA* is the northern hemisphere of the event horizon globe if the luminous accretion disk around this black hole is thin. The dark spot is much smaller of the classical black hole shadow. The outer boundary of this dark spot is an equator on the event horizon globe.

I believe that this paper is interesting and useful for the potential readers. For the benefit of the readers I would request some minor changes.

Minor requests:

1. Problem with order of calling references in section Introduction should be repaired: [7,24] [9–20].

2. Calling the reference [8] from reference list: (Chandrasekhar, S. Chapter 7. The Mathematical Theory of Black Holes. ...) is not appear in the text.

3. It should be mentioned paper Falcke et al_2000_ApJ_528_L13 and results of their simulations and predictions.

I would like to recommend publication of this paper after minor revision.

Author Response

Point 1: Minor requests:
1. Problem with order of calling references in section Introduction should be repaired: [7,24] [9–20].
2. Calling the reference [8] from reference list: (Chandrasekhar, S. Chapter 7. The Mathematical Theory of Black Holes. ...) is not appear in the text.
3. It should be mentioned paper Falcke et al_2000_ApJ_528_L13 and results of their simulations and predictions.

Response1: Thanks, all references are corrected. Reference to Falcke et al. is added.

Reviewer 3 Report

The authors claim provided the definitive evidence that the dark spot at the image of SgrA* obtained by the EHT collaboration, and is due to is the northern hemisphere of the event horizon globe if the luminous accretion disk around this black hole is thin.
I don't find any novelty in this paper, any deep and relevant explications. Moreover, its majority is derived from the Ref[12].
It's obvious that this work doesn't match the criteria for publication in such a high-level journal. Thus I'm in the obligation to reject it.

Author Response

Response to Reviewer 3 Comments

Point 1: Comments and Suggestions for Authors

The authors claim provided the definitive evidence that the dark spot at the image of SgrA* obtained by the EHT collaboration and is due to is the northern hemisphere of the event horizon globe if the luminous accretion disk around this black hole is thin.

I don't find any novelty in this paper, any deep and relevant explications.

Response 1: The revised note version is extended and proofed. In the submitted revision of this note all new inclusions are marked by magenta color.

Reviewer 4 Report

The article deals with the recent evidence that supports the existence of black holes, the image of the supermassive black hole Sagittarius A. The author uses Kerr's metric to model the physical system in question. The article is very interesting and has a lot of appeal to the community. However, I think the discussion about the image should be improved. Questions how it was obtained and what its limitations should be answered.

Author Response

Response to Reviewer 4 Comments

Point 1: I think the discussion about the image should be improved. Questions how it was obtained and what its limitations should be answered.

Response 1: The revised note version is extended and proofed. In the submitted revision of this note all new inclusions are marked by purple color.

Round 2

Reviewer 1 Report

Point 1: It is claimed in the introduction that the distinctive dark spot is smaller than the expected size. However, the size of the dark spot depends on the angular resolution, which is lowered by the variability of the source. This aspect is entirely ignored by the analysis reported in this paper.

Response 1: Angular resolution of the first SgrA* is very poor. For this reason the using of the transparent thin disk model in this note (or any other) is justified.

Reply: This response does not address the issue of the dependence of the dark spot on the angular resolution at all. 

Point 2: The conclusions of this paper are not clear to me. In fact, with reference to Fig.~1, the purple curve does not match the dark spot.

Response 2: The purple curve does not match the dark spot because it is the classical black hole shadow, which is invisible in this image. This purple curve matches the luminous inner part of the accretion disk near the outer boundary of this purple curve. Instead of, namely the dark silhouette of the northern hemisphere of the black hole event horizon matches the rather small dark spot rather well (if black hole spin >0.5). In the revised note the Fig.1 is improved for better coincidence of the dark spot with our dark spot mode (by remembering the pure EHT angular resolution and 3-sigma uncertainty).

Reply: I would like to thank the author for the clarification. This point is actually more clear now. However, it is still not clear why it should reasonable that only the northern hemisphere casts the shadow. Furthermore there is not enough evidence that the match is improved as the visual representation is not significative if not accompanied by an analysis on the resolution and on the sensitivity.

Point 3: Furthermore, it is explained that the physical origin of the dark spot has been obtained ``by using numerical calculation of photon trajectories, starting a little bit above the black hole event horizon and reaching a distant static observer just above the black hole equatorial plane". This procedure is incorrect. In fact, most of the matter inside the photon sphere will not escape to infinity. This implies that the dark spot is actually associated with the photon sphere, not the horizon.

Response 3: There is no any need for the matter inside the photon sphere to escape to infinity to provide the observable black hole image. Only photons from the lumunous accretion disk in the vicinity of the black hole event horizon may happily escape to a distant observer (with the correponding Doppler and gravitational energy shifts). It is verified in numerous published simulations and in our previous published papers.

Reply: This is the main problem of the paper. Photons in the vicinity of the horizon do not escape to a distant observer. Sure, there exist a geodesic from arbitrary close to the horizon that reaches the distant observer, however the vast majority of geodesics too close to the horizon end up into the black hole. Therefore the region between the photosphere and the horizon is not luminous enough to be observable. It is the photonsphere (which is bigger than the horizon) that casts the shadow. 

Point 4: Let me also stress that the paper assumes 
1. Thin accretion disk;
2. Rotation axis of SgrA* coincides with a rotation axis of the Milky Way galaxy.
Neither of these assumptions is justified in the text.  

Response 4: In the revised note we state that ``thin accretion disk'' is a stated model in our note. It is a very natural idea (hypothesis) that ``rotation axis of SgrA* coincides with a rotation axis of the Milky Way galaxy'' proposed and used in many publications.

Reply: From the analysis of the EHT collaboration it is clear that this hypotheses, while not excluded, are not particularly favoured by the data. Therefore, I could accept them as simplifying assumptions, but only if clearly stated and only if the result would not depend strongly on them. 

Point 5: Finally, let me point out a minor problem. The horizon is always called ``event horizon" rather than ``apparent horizon". This terminology is very common (also in the name of the EHT collaboration), but it is wrong anyway. The event horizon has a precise mathematical definition and it is not observable [arXiv:1407.7295].

Response 5: Yes, the event horizon of the black hole is not observable. However, the observable is the ``reconstructed’’ dark silhouette of the event horizon. Reconstruction means that it is possible to observe photons emitted be the infalling accretion matter in the vicinity of the event horizon. The last emitted photons and detected by a distant telescope mark the outer boundary of the dark spot at the black hole image. In the thin accretion disk model this outer boundary of the dark spot is the ``reconstructed’’ image of the equator at the black hole event horizon globe. In the thin accretion disk model the dark spot in the case of the SgrA* is the northern hemisphere at the black hole event horizon globe (while in the case of M87* the dark spot is the southern hemisphere at the black hole event horizon globe.). This clarification is added to Introduction of the updated note.

Reply: Maybe I was not particularly clear in this point. I apologise if that is the case. What I meant is that the event horizon has a precise mathematical definition which is a global definition (meaning that depends on the entire future history of the universe). Therefore it is not a physically observable quantity (even with infinite precision). What would be obserble (if we had infinite precision) is the trapping or apparent horizon which have a quasi-local definition. In the static geometry trapping and event horizon are at the same location, but for an astrophical black hole which is acrreting matter the two horizons are separated. This issue is explained very well in the paper I quoted. As I said, this is a minor issue and it is only about the naming we give to a certain object. I am ok if the author (along the line of the majority of the works in the astrophysics community) wants to name the apparent horizon ``event horizon" What is very important, is that the reply of the author ``...However, the observable is the ``reconstructed’’ dark silhouette of the event horizon. Reconstruction means that it is possible to observe photons emitted be the infalling accretion matter in the vicinity of the event horizon. The last emitted photons and detected by a distant telescope mark the outer boundary of the dark spot at the black hole image" is simply wrong. As I explained above, we can at best reconstruct the siluette of the photonsphere. 

In conclusion, I strongly beleive that this paper is not suitably for publication.

Author Response

Response to Reviewer 1 Round 2 Comments

Point 1: It is claimed in the introduction that the distinctive dark spot is smaller than the expected size. However, the size of the dark spot depends on the angular resolution, which is lowered by the variability of the source. This aspect is entirely ignored by the analysis reported in this paper.

Response 1: Angular resolution of the first SgrA* is very poor. For this reason, the using of the transparent thin disk model in this note (or any other) is justified.

Reply: This response does not address the issue of the dependence of the dark spot on the angular resolution at all.

Response 1 Round2: Angular resolution of the first SgrA* is very roughly 5microarcsec, the size of the shadow (our purple closed curve) is roughly 50microarcsec. But the size of the dark spot is roughly 10microarcsec. The visible size of the dark spot is reasonably and sufficiently smaller the corresponding one of the shadow! This is the main observation in this my note. Moreover, the small size of the dark spot is in a good agreement with the thin accretion disk model

Point 2: The conclusions of this paper are not clear to me. In fact, with reference to Fig.~1, the purple curve does not match the dark spot.

Response 2: The purple curve does not match the dark spot because it is the classical black hole shadow, which is invisible in this image. This purple curve matches the luminous inner part of the accretion disk near the outer boundary of this purple curve. Instead of, namely the dark silhouette of the northern hemisphere of the black hole event horizon matches the rather small dark spot rather well (if black hole spin >0.5). In the revised note the Fig.1 is improved for better coincidence of the dark spot with our dark spot mode (by remembering the pure EHT angular resolution and 3-sigma uncertainty).

Point 2: The conclusions of this paper are not clear to me. In fact, with reference to Fig.~1, the purple curve does not match the dark spot.

Response 2: The purple curve does not match the dark spot because it is the classical black hole shadow, which is invisible in this image. This purple curve matches the luminous inner part of the accretion disk near the outer boundary of this purple curve. Instead of, namely the dark silhouette of the northern hemisphere of the black hole event horizon matches the rather small dark spot rather well (if black hole spin >0.5). In the revised note the Fig.1 is improved for better coincidence of the dark spot with our dark spot mode (by remembering the pure EHT angular resolution and 3-sigma uncertainty).

Reply: I would like to thank the author for the clarification. This point is actually more clear now. However, it is still not clear why it should reasonable that only the northern hemisphere casts the shadow. Furthermore, there is not enough evidence that the match is improved as the visual representation is not significative if not accompanied by an analysis on the resolution and on the sensitivity.

Response 2 Round2: The northern hemisphere casts the shadow due to the natural position of the distant telescope (EHT) a little bit above the equatorial plane of the SgrA* (of course, additionally it is supposed that the rotation axis of the SgrA* coincides with the rotation axis of the Milky way). This is explained in detailes in our cited publications.

Point 3: Furthermore, it is explained that the physical origin of the dark spot has been obtained ``by using numerical calculation of photon trajectories, starting a little bit above the black hole event horizon and reaching a distant static observer just above the black hole equatorial plane". This procedure is incorrect. In fact, most of the matter inside the photon sphere will not escape to infinity. This implies that the dark spot is associated with the photon sphere, not the horizon.

Response 3: There is no any need for the matter inside the photon sphere to escape to infinity to provide the observable black hole image. Only photons from the lumunous accretion disk in the vicinity of the black hole event horizon may happily escape to a distant observer (with the corresponding Doppler and gravitational energy shifts). It is verified in numerous published simulations and in our previous published papers.

Reply: This is the main problem of the paper. Photons in the vicinity of the horizon do not escape to a distant observer. Sure, there exist a geodesic from arbitrary close to the horizon that reaches the distant observer, however the vast majority of geodesics too close to the horizon end up into the black hole. Therefore, the region between the photosphere and the horizon is not luminous enough to be observable. It is the photonsphere (which is bigger than the horizon) that casts the shadow.

Response 3 Round2: The statement that ``the region between the photosphere and the horizon is not luminous enough to be observable'' is very superficial and model dependent. Quit in opposite, the recent GRMHD simulations demonstrate that in the framework of the Blandford-Znajek mechanism the observable accretion disk (heated by the lectric current (flowing through the disk and black hole horizon) is observable even in the vicinity of the event horizon due to the very high accretion luminosity (see details in [22,23]).

Point 4: Let me also stress that the paper assumes

  1. Thin accretion disk;
  2. Rotation axis of SgrA* coincides with a rotation axis of the Milky Way galaxy.

Neither of these assumptions is justified in the text. 

Response 4: In the revised note we state that ``thin accretion disk'' is a stated model in our note. It is a very natural idea (hypothesis) that ``rotation axis of SgrA* coincides with a rotation axis of the Milky Way galaxy'' proposed and used in many publications.

Reply: From the analysis of the EHT collaboration it is clear that this hypothesis, while not excluded, are not particularly favoured by the data. Therefore, I could accept them as simplifying assumptions, but only if clearly stated and only if the result would not depend strongly on them.

Response 4 Round2: Yes, this is simplifying assumption. But this assumption is very natural!

Point 5: Finally, let me point out a minor problem. The horizon is always called ``event horizon" rather than ``apparent horizon". This terminology is very common (also in the name of the EHT collaboration), but it is wrong anyway. The event horizon has a precise mathematical definition, and it is not observable [arXiv:1407.7295].

Response 5: Yes, the event horizon of the black hole is not observable. However, the observable is the ``reconstructed’’ dark silhouette of the event horizon. Reconstruction means that it is possible to observe photons emitted be the infalling accretion matter in the vicinity of the event horizon. The last emitted photons and detected by a distant telescope mark the outer boundary of the dark spot at the black hole image. In the thin accretion disk model this outer boundary of the dark spot is the ``reconstructed’’ image of the equator at the black hole event horizon globe. In the thin accretion disk model, the dark spot in the case of the SgrA* is the northern hemisphere at the black hole event horizon globe (while in the case of M87* the dark spot is the southern hemisphere at the black hole event horizon globe.). This clarification is added to Introduction of the updated note.

Reply: Maybe I was not particularly clear in this point. I apologise if that is the case. What I meant is that the event horizon has a precise mathematical definition which is a global definition (meaning that depends on the entire future history of the universe). Therefore, it is not a physically observable quantity (even with infinite precision). What would be observable (if we had infinite precision) is the trapping or apparent horizon which have a quasi-local definition. In the static geometry trapping and event horizon are at the same location, but for an astrophysical black hole which is accreting matter the two horizons are separated. This issue is explained very well in the paper I quoted. As I said, this is a minor issue and it is only about the naming we give to a certain object. I am ok if the author (along the line of the majority of the works in the astrophysics community) wants to name the apparent horizon ``event horizon" What is very important, is that the reply of the author ``...However, the observable is the ``reconstructed’’ dark silhouette of the event horizon. Reconstruction means that it is possible to observe photons emitted be the infalling accretion matter in the vicinity of the event horizon. The last emitted photons and detected by a distant telescope mark the outer boundary of the dark spot at the black hole image" is simply wrong. As I explained above, we can at best reconstruct the silhouette of the photonsphere.

Response 5 Round2: Besides the photons, emitted near the photonsphere (and not registered by EHT due to the low brightness) there are photons emitted inside the photonsphere but above the event horizon, which can escape to the distant telescope (of course with Doppler and gravitational energy shifts). Namely these photons provide the possibility for the the event horizon silhouette ``reconstruction’’. This ``reconstruction’’ is explained in detailes in our previous publications (cited in this note).

Reviewer 3 Report

The paper can be published now

Author Response

I thank the Reviewer 3 for his positive decision.